# Mobile and Online Health Information: Exploring Digital Media Use among Austrian Parents

**DOI:** 10.3390/ijerph17176053

**Published:** 2020-08-20

**Authors:** Daniela Haluza, Isabella Böhm

**Affiliations:** Department of Environmental Health, Center for Public Health, Medical University of Vienna, 1090 Vienna, Austria; isabella.nineol@hotmail.com

**Keywords:** pediatrics, health information technology, health seeking behavior, influencing factors, online resources

## Abstract

In today’s digitalized world, most parents are Internet-savvy and use online sources for child health information, mainly due to the 24/7 availability of advice. However, parents are often not specifically trained to identify reliable, evidence-based sources of information. In this cross-sectional online survey among a purposive, non-probabilistic sample of Austrian parents (*n* = 90, 81.1% females), we assessed aspects of health app use and family policy benefits-related and scenario-based Internet seeking behavior. We found that the surveyed parents showed a high health app use. The participants indicated that they prefer online information seeking to any other option in a scenario describing that their child would be sick at after-work hours, with social media channels being the least preferred source of online information. Mothers and younger parents were more likely to retrieve online information on family policy benefits. With the smartphone in everybody’s pocket, parents seemed to rely on mobile and online content when searching for child health information. Pediatricians are best suited to decide what treatment fits the child or their current medical condition, but nowadays they face increasing numbers of pre-informed parents seeking health information online. Provision of targeted parental education and guidance through the online information jungle could effectively empower parents and smooth personal and digital contacts in the delicate doctor–parent–child triangle.

## 1. Introduction

In today’s digitalized society, the Internet has become a highly frequented source of health information. Web sources are already more popular for health information retrieval than healthcare professionals [1]. Demographic features influence online habits in general, with gender being the most frequently studied personal characteristic in this respect with a female dominance in, e.g., seeking online health information and using health apps [2,3,4,5,6,7,8]. Apps for tracking activity and calorie intake are the most popular [9,10,11].

An unprecedented majority of parents use Web-based sources for actively seeking advice on child health and development issues, with this pre-obtained information increasingly influencing the per se delicate interaction in the doctor–parent–child triangle [1]. However, research suggests that for the sake of the child’s well-being, online information is in most cases used in addition to—and not in replacement of—personal contact with a healthcare facility [12]. General Internet use of parents is independent of their gender and educational level, but younger parents and parents of younger children are more prone to retrieve online information regarding child health [13]. Already in 2015, 98% of Canadian parents sought online content [14]. Jaks et al. found in a 2019 study that 91% of Swiss–German parents use these sources [12].

Mainly by using general search engines and parental webpages, parents search for advice regarding common health conditions, such as fever, colds, and skin rashes. Largely, digital media is perceived as a convenient resource, especially by fathers. However, parents also worry about the correctness of online content and whether their interpretive understanding is adequate, both of which reduces their confidence in derived child care actions. As a result, two-thirds of the parents in the Swiss–German study asked for further help from a pediatrician, and half of those in the Canadian study discussed the retrieved content with a healthcare professional [12,14]. Van der Gugten et al. found that Internet information—although predominantly used for this purpose—does not influence decision-making among Dutch parents [15]. An Austrian study from 2016 found that one fifth of the parents attending an outpatient clinic used digital media prior to the appointment [16]. Higher education level of parents as well as young age and acute illness of their child predisposed them to online health information seeking. This study did not find evidence that online health information retrieval triggers panic among parents when searching for online content regarding child health [16]. To date, little is known about how Austrian parents in general use digital media to engage in the topic of child health. In this exploratory study on digital media use, we studied a convenience sample of parents to assess health app use, use of online sources, and which hypothetical scenarios would come into effect in the case of a health emergency in the context of childcare.

This questionnaire-based online study aimed at evaluating parental degree of readiness for using health apps using the commonly used readiness assessment for telehealth tool (PRAT) [17,18]. Originally in English, the PRAT collects opinions in three categories: core readiness covers dissatisfaction with the status quo, expectation of change, and desire to use health apps; engagement readiness covers awareness and perceived health app use-related benefits and barriers; structural readiness covers availability of health app use-related technical infrastructure and soft skills. The resulting readiness score indicates health app use readiness, i.e., whether the participant is in a good position or experiences certain barriers to health app usage.

## 2. Methods

### 2.1. Study Design

We conducted a Web-based questionnaire study among a cross-section of the Austrian population. We developed a German study questionnaire based on previous surveys and details are reported elsewhere [1,10,18,19,20]. The online survey was enabled by the Web-based survey tool SoSci Survey and was accessible barrier-free from 28 February to 14 June 2018 [21]. Attempting a snowball sampling approach, we sent the non-personalized link to the online questionnaire to potential survey participants via email invitations and posted it to social media sites and groups (Facebook, WhatsApp) as well as specific parenting blogs, with the request to circulate the link in the respondents’ own network of friends and family [22,23]. The cover page informed participants about the study aim, and their consent was implicitly obtained when completing the survey. The survey protocol was approved by the institutional ethical committee of the Medical University of Vienna, Austria, on 1 February 2018 (no. 0102/2018). The study was conducted following the ethical standards laid out in the Declaration of Helsinki.

The link to the online survey was accessed 1605 times; 324 participants started and 217 of those completed the survey (66.98% completion rate). The questionnaire collected socio-demographic data such as age, gender, education, place of residence, having children, age of children (youngest and oldest one), and smartphone use, as well as aspects of health app use (yes or no) and frequency (daily, several times a week, weekly, monthly or less than once a month). We used a filter question (i.e., having children) to identify parents who then exclusively filled out parenting-specific questions. We thus excluded 127 cases from the final dataset, which then included in total 90 study subjects—73 females and 17 males. Participants needed in average 7.43 min (SD 1.83 min) to completely fill out the questionnaire.

### 2.2. Study Questionnaire

Besides the socio-demographic characteristics mentioned above, participants were asked to rate seven potential benefits of health app use in general on a 5-point Likert scale (1 = strongly agree to 5 = strongly disagree). Choices referred to benefits regarding location-independent healthcare access, efficiency in healthcare resource allocation, quality of healthcare, healthcare costs, efficiency in medical consultation, multiple diagnostics, and doctor–patient relationship. These items formed the total benefits score.

A multiple-choice question asked participants to indicate whether they had already searched online for information on family policy benefits (yes vs. no). Potential choices included information on family benefits, childcare benefits, parental leave, sole earner deduction, parental part-time work, consultation, and parental education, adapted from the official categories of the Austrian Federal Ministry for Families and Youth [24].

The further survey part contained the adapted version of the patient/public PRAT [17,18,25,26]. This assessment tool is available in three versions tailored at the three participant groups, namely, patient/public, healthcare practitioners, and organizations. The original English version of the PRAT assessed readiness for telehealth, which we replaced with “health app use” in our questionnaire. This procedure is highly encouraged by the developers of the original PRAT.

In the present study, we translated and adapted the patient/public tool using translation and back-translation procedures to develop a German study questionnaire [1,10,18,20,27]. We pre-tested the questionnaire with eleven participants to ensure general comprehensibility, face validity, and content validity. We integrated the received feedback in the final survey. In the pre-test, two items from the original PRAT stood out as irrelevant for assessing health app use, namely “benefits or anticipated benefits/risks” and “practitioner-mediated liaison for telehealth programs”. Thus, we excluded these two items from the tool used in this study, resulting in a final scale consisting of 15 items using a 5-point Likert scale (1 = strongly agree to 5 = strongly disagree). These 15 items formed the total readiness score.

#### Scenarios

Two scenarios surveyed preferred sources for child health information (scenario 1), and, in cases of Internet searches, which platforms were used (scenario 2). We developed the scenarios for anticipating real-life situations of participants, not best-case and worst-case scenarios, and each of the two scenarios was consistent. Integrating existing knowledge from the pertinent literature, the scenarios simplified a complex picture to more clearly outline the inherent meaning [1,12,13,14,15,16]. Scenario 1 describes the following situation: “Imagine that at 8:00 p.m. your child suddenly shows an itchy rash on the body and complains about severe headache. The private practices of your general practitioner and your pediatrician are already closed. You do not know why your child feels that way. What would you do?”. Participants were asked to rank the following five options by probability: look for the symptoms on the Internet, ask friends or relatives for advice, consult a pediatric book, drive the child to the nearest hospital immediately, and call an ambulance. In the pre-test, we explicitly verified that in the case of an emergency, people in Austria would commonly not hesitate to call their closest friends or relatives at any time (even at midnight). Thus, we chose the time of 8 p.m. for convenience reasons, as at this time, doctors’ offices are already closed, but most people are still awake.

Scenario 2 describes the following situation: “Imagine that at 8:00 p.m. your child suddenly shows an itchy rash on the body and complains about severe headache. The private practices of your general practitioner and your pediatrician are already closed. You do not know why your child feels that way. You want to visit your family doctor in the morning if your child does not feel better by then, but you also want to search online about your child’s condition.” Participants were asked for the likelihood of using distinct online sources on a 4-point Likert scale ranging from low = 1 to high = 4. Assuming that most participants would indicate using a general search engine, e.g., Google [14], the specific sources were Netdoktor [28], an Internet forum, a homepage of a federal ministry, Gesundpedia [29], DocCheck Flexikon [30], and a social media channel.

### 2.3. Statistical Data Analysis

We conducted all reported statistical analyses using SPSS Statistics for Windows, Version 26.0 (IBM Corp, Armonk, NY, USA). We calculated the aggregated scores, namely benefits score (i.e., mean score of the items of benefits of health apps) and readiness score (i.e., mean score of the 15 items of the PRAT), with lower ratings indicating higher degree of agreement [25]. To determine the internal consistency of the scores, we calculated Cronbach’s alpha, with a threshold of 0.6 for acceptable values [31,32]. The benefits score showed a good internal consistency of alpha = 0.83 and the readiness score an acceptable internal consistency of alpha = 0.65.

Median splits created dichotomized variables for age (young vs. older), i.e., a young age group (*n* = 44, younger than 40 years, and an older age group (*n* = 46 years and older). We descriptively summarized collected data to present categorical data as absolute and relative frequencies, and continuous data as mean, standard deviation (SD). We determined subgroup differences for age (young vs. older) and gender (male vs. female) using chi^2^ tests (regarding health app use and family policy benefits) and Mann–Whitney U tests (regarding benefits score and readiness score).

## 3. Results

Most of the study subjects (*n* = 90, 81.1% females) had one (43.3%) child, 37.8% had two, 13.3% had three, and 5.6% had four children. The average age of study participants was 42.19 (SD 10.77, range 23–72 years). The average age of their children was 7.16 years (SD 7.52, range 1–20 years). Median split resulted in a younger and an older group (*n* = 47, 52.2%, *n* = 43, 47.8%, respectively). Most participants lived in Lower Austria (44.4%) and Vienna (35.6%). More than half of the participants (56.7%) reported to have a primary and secondary education, whereas 43.3% had a higher education.

In our study sample, health app use was high (96.7%). Health apps were used quite regularly, with 19.4% reporting a daily, 30.6% a several times a week and another 30.6% a weekly use. The most popular health apps were those for monitoring physical activity (34.4%), eating habits, weight reduction, and quality of sleep (all: 12.2%). We did not find statistically significant age group and gender differences (chi^2^ test, all *p* > 0.05).

As for family policy benefits, we found that about 70% of participants searched the Web for information on family benefits (73.3%), childcare benefits (72.2%), and parental leave (70.0%), about half of the study subjects searched for information on sole earner deduction and parental part-time work (both 51.1%) as well as consultation (46.7%), whereas 25.6% of participants searched for information on parent education. With females showing more interest in online information, we found statistically significant gender differences in chi^2^ tests for family benefits (chi^2^ (1, *n* = 90) = 11.1, *p* < 0.001), childcare benefits (chi^2^ = 3.9, *p* = 0.049), parental leave (chi^2^ = 5.3, *p* = 0.022), and parent education (chi^2^ = 4.3, *p* = 0.039). With younger participants showing more interest in online information, we found statistically significant age group differences for family benefits (chi^2^ = 5.1, *p* = 0.024), childcare benefits (chi^2^ = 18.9, *p* < 0.001), parental leave (chi^2^ = 17.9, *p* < 0.001), and parental part-time work (chi^2^ = 19.7, *p* < 0.001).

For participants, the top benefit of health apps was “location-independent access to health services” (mean 2.32, SD 1.11), whereas “improved doctor–patient relationship” yielded the lowest approval (mean 3.44, SD 0.94, Table 1). The benefits score was on average 2.9 (SD 0.71), showing statistically significant gender differences, as females were less likely to be in favor of health apps compared to males (mean 3.0 SD 0.7 vs. mean 2.4 SD 0.6, *p* from Mann–Whitney U test = 0.003), whereas age groups did not differ in this respect.

Table 2 shows the ratings for the readiness assessment tool. The overall readiness score was on average 2.92 (SD 0.45, 15 items). In the domain core readiness, the item “I have a desire for change and want to actively be involved in my health and health care condition” scored highest (mean 2.46, SD 1.22). In engagement readiness, the item “I have a sense of ownership regarding my wellbeing and that of my community” scored highest (mean 1.22, SD 0.58). In structural readiness, the item “I have access to health apps and the ability to use them” scored highest (mean 1.62, SD 0.87). We found the overall lowest approval for “I am aware of education campaigns about health apps” (mean 4.48, SD 0.86). We did not find statistically significant subgroup differences (age and gender) for the readiness score (Mann–Whitney U test, all *p* > 0.05), and thus abstained from reporting *p* values in Table 2.

Scenario-based evaluations are depicted in Table 3. Scenario 1 analyzed participants’ first anticipated actions to evaluate their child´s health condition in a ranking question (rank 1–5). Before seeking medical support, participants indicated consulting online sources, friends or relatives, or a pediatric book. Scenario 2 specifically asked for the likelihood to use distinct online platforms (low = 1 to high = 4). The well-known German webpage Netdoktor [28] ranked highest (mean 3.80, SD 0.85), whereas a social media channel ranked lowest (mean 1.54, SD 0.77). The ranks 2 to 5 were covered by an Internet forum, the homepage of a federal ministry, Gesundpedia [29], and DocCheck Flexikon [30]. While genders did not differ in this respect according to Mann–Whitney U tests, the young age group was less likely to ask friends or relatives for advice (mean 2.0 SD 1.1 vs. mean 2.8 SD 1.3, U = 691, *p* = 0.007), but more likely to consult a pediatric book (mean 3.7 SD 1.3 vs. mean 2.8 SD 1.4, U = 653, *p* = 0.003) in scenario 1, and seek advice from an Internet forum (mean 2.3 SD 0.8 vs. mean 1.9 SD 0.8, U = 727, *p* = 0.015) in scenario 2, respectively.

## 4. Discussion

The present study aimed at understanding the prevailing preferences of parents and how preferences varied by age and gender in the context of digital media usage. In a cross-sectional design, we assessed parental health app use and online information retrieval among a purposive sample of Austrian participants. Interestingly, health app use was considerably high, but perceived benefits and readiness to use them ranged only in a medium range. The reason could be that users perceived health apps as means for private communication and data collection independent from medical support and healthcare services. This is supported by the high ratings for the aspect of location-independent access to health services, a very obvious and inherent benefit of all digital devices enabling telehealth, as well as the low ratings for doctor–patient relationship improvement [33]. In line with previously published studies, participants predominantly used health apps for tracking physical activity and calorie consumption [9,10,11].

Scenario-based assessments are used in a variety of research areas, for example to capture public views and preferences on healthcare compensation [34], preferences of patients in regard to IT-assisted diabetes care [35], implementation of digital healthcare [36], and perceptions of medical app use in clinical communication [37]. In this methodological approach, we asked questions based on scenarios rather than exploring or observing actual interactions. In doing so, we could standardize the context using vignettes similar to real-world experiences, elucidating potential real-life actions in need for preferably fast solutions. In line with the prospect theory, we thus asked for a “doing something” decision, which is commonly perceived as less risky than doing nothing [38]. Our study focused on parents, so the scenarios are presumably not relevant for those not guarding children. The scenarios depicting hypothetical situations showed interesting findings. First, the participants indicated that they prefer online information seeking to any other action in our first vignette describing real events, and second, social media channels are the least preferred source when doing so in scenario 2. In addition, younger parents were less likely to ask friends and relatives, but were more likely to consult a pediatric book. This observation could eventually describe a conditional information retrieval behavior combining the existence of established and reliable social networks higher among older parents with availability of up-to-date pediatric literature and higher health literacy of younger adults. If someone has already sought advice from friends and relatives who are very likely also parents themselves, he or she will be less likely to pick up a book which was bought many years ago.

It is assumed that especially in the health context, a very large part of online material is written in language too complex for the majority of the general public to understand, let alone act on it in a stressful situation. A representative survey among Austrian adults on the prevailing knowledge and evaluation of the offered family benefits found that more than half of the respondents found it difficult to retrieve useful information about family policy benefits [24]. Most parents take advantage of cash benefits and maternity leave, but hardly any parental part-time work is claimed and less than 10% use advisory services. In our study, we found a high affinity to retrieve family-specific information from online sources among the surveyed, definitely Internet-savvy parents, with information on family and childcare benefits and parental leave being the most popular. These search terms inherently claim quite complex legal frameworks that are of utmost importance to young parents or parents to be, as they are entitled to considerable financial benefits [24]. Additionally, the options and requirements of family and childcare benefits and parental leave have been due to some changes in the last couple of years following the establishment of a new government December 2017, shortly before this survey was conducted, often leaving parents to actively seek advice on the Web for initial guidance.

Pehora et al. found that 80% of the surveyed parents start their search on a general search engine such as Google and Wikipedia [14]. Despite rather low use of health institutional websites of about one-fifth, three-quarters of respondents regarded the content provided there as safe and reliable, which was the case for only about 25% for the general search engines. Increase in health literacy and empowerment of parents is an important, but neglected and underfinanced aspect in strenuous point of care and counseling settings. The lack of adequate funding for pediatricians and other healthcare professionals to date is a prerequisite for time-consuming counseling in the doctor–parent–child triangle. As a potential starting-point, Megan Moreno [39] suggests a simple four-step strategy for parents searching for information that allows them to evaluate the information to be sure that it is accurate and trustworthy. Search: start your search on an institutional health-focused website instead of Google or other general search engines. Evaluate: identify the author´s credentials and evaluate the provided references. Check: check at least three other online sources for similar information. Ask: consult your pediatrician for advice and discuss the health information retrieved online. In the case that parents cross-check the information retrieved online with a healthcare professional, best with a pediatrician, this easy four-step instruction could be a useful point of care checklist in doctor–parent counseling situations for improving outcomes of online searches.

Our study participants showed a heterogeneous background with a medium average age of 42 years with one to four children and a high percentage of females (81%). Both health app use and education level was above average, indicating a stratum of the population with likelihood to be targeted by parental education endeavors. A somewhat similar study population profile was reported in a Swiss study on parents health literacy with almost 90% females and an outstandingly high socioeconomic status (77% highly educated participants) [40]. There is evidence that especially first-time parents might differ regarding their need for professional advice and their use of the out-of-hours services in childcare compared to more experienced parents [41]. Indeed, we found that younger participants showed more interest in online information, also due to a higher affinity to digital media in so-called digital natives [1].

## 5. Limitations

Our study had limitations that should be kept in mind when interpreting our observations. We aimed at analyzing the current status of health app use and assessing readiness for health app use among a cross-section of Austrian parents reachable by an online snowball sampling approach, limiting generalizability of study results and representativeness of the study sample. As with all cross-sectional surveys, this was a study of association, not causation, and we were not able to evaluate trends in individual behavior over time. There is the possibility of unmeasured confounding associated with app use and intentions or health behaviors, which could influence the interpretation of results. Notably, we did not conduct detailed subgroup and multivariate analyses (correlations or multiple regressions) that would go beyond the scope of the study.

This study could not answer the question of whether per se more motivated individuals sought out apps, or whether app use improved their motivation and subsequently their health. We did not assess details about whether the apps were interactive and linked to health promotion support such as telehealth and other strategies for health behavior change.

We excluded parents who did not read and understand German. The patterns of Internet use might be influenced by socio-cultural characteristics including digital age and also language, something our study did not investigate. Finally, we relied on participants to self-report sources they accessed, which introduced recall bias. As for the scenarios, we assumed that participants judged scenario validity based on the level of scenario realism rather than on its ability to answer predefined questions.

As we conducted an online exploratory study, we abstained from sample size calculation and did not use advanced recruiting strategies and incentives for participation, which could clearly increase response rates in follow-up studies [22]. Nevertheless, similar sample sizes are also reported in related studies [8,18,20,42]. As expected, the age of parents correlated with the age of the children. Parents with more than one child might also have more confidence when a younger sibling gets sick. On the other hand, parents with a chronically ill child might act differently. Thus, aspects of child age and other child-specific characteristics might be integrated in further studies. These limitations non withstanding, the survey tool used in this study and the reported findings should be refined in more complex research approaches, e.g., mixed-method approaches, in a large, representative study population to elucidate further mechanisms of information retrieval among German-speaking parents and guardians.

## 6. Conclusions

This study on digital media in childcare increases the understanding of parental child health information retrieval preferences and how preferences varied by age and gender. In today’s digitalized world, parents more and more rely on Web content when searching for child health information. With the smartphone in everybody’s pocket, this trend is very likely to be continued in the future. Pediatricians are trusted experts in giving evidence-based advice on childcare and treatment options, but an increasing numbers of pre-informed parents at point of care affect the traditional doctor–patient relationship in the digital age. Thus, mutual understanding and in-depth knowledge of information retrieval habits in general and specifically in the target group are essential in the per se delicate and complex interactions expectable in the doctor–parent–child triangle. For this, legal support and pro-family policies should integrate this newly established Web-based parental empowerment and health literacy in family and health policy and decision-making processes.

## Figures and Tables

**Table 1 ijerph-17-06053-t001:** Benefits of health apps (*n* = 90).

Items	Mean	SD	*p* #	
			Age	Gender
Location-independent access to health services	2.32	1.11	0.929	0.076
Higher efficiency in healthcare resource allocation	2.79	1.20	0.440	0.003
Higher quality of healthcare	2.86	1.05	0.844	0.072
Reduced healthcare costs	2.93	1.02	0.359	0.001
Higher efficiency in medical consultation	3.11	1.04	0.204	0.004
Reduced multiple diagnostics	3.13	1.20	0.164	0.006
Improved doctor–patient relationship	3.44	0.94	0.741	0.306
**Benefits Score**	2.86	0.71	0.485	0.003

Notes: # Mann–Whitney U tests for the dichotomous variables age groups (young vs. older) and gender (males vs. females).

**Table 2 ijerph-17-06053-t002:** Public readiness assessment for health app use (*n* = 90).

As a Person, in Order to Meet the Requirements for Health App Use, I…:	Mean	SD
**Core readiness**		
Feel dissatisfied with usual doctor–patient interaction or have a desire for a more comfortable setting for obtaining health information.	3.53	1.42
Identify with a sense of dissatisfaction with the current state of healthcare.	3.71	1.28
Identify with a sense of isolation and a lack of access to healthcare.	4.12	0.96
Acknowledge unmet healthcare needs.	3.92	1.27
Have a desire for change and want to actively be involved in my health and healthcare condition.	2.46	1.22
**Engagement readiness**		
Believe that health apps are not a replacement, but an addition to traditional care.	1.59	0.85
Believe that cultural issues can be addressed when using health apps.	2.70	1.03
Believe that concerns specific to privacy/confidentiality/security have been addressed when using health apps.	3.04	1.20
Have a sense of ownership regarding my wellbeing and that of my community.	1.22	0.58
Am comfortable with using health apps.	2.13	1.19
Am knowledgeable about health apps and/or want to know what health apps are.	1.76	0.90
**Structural readiness**		
Have access to information about health apps from official sources (e.g., brochures, from doctors).	2.83	1.28
Have access to health apps and the ability to use them.	1.62	0.87
Am aware of education campaigns about health apps.	4.48	0.86
Am a local champion who has an ambition to bring telehealth to my community.	3.62	1.12
**Readiness score**	2.92	0.45

**Table 3 ijerph-17-06053-t003:** Ratings of scenarios 1 and 2 (*n* = 90).

	Mean	SD	*p* #	
			Age	Gender
**Scenario 1:**				
**Imagine that at 8:00 p.m. your child suddenly shows an itchy rash on the body and complains about severe headache. The private practices of your general practitioner and your pediatrician are already closed. You do not know why your child feels that way.** What would you do? Please rank the following options from 1–5.				
I look for the symptoms on the Internet.	2.20	1.26	0.280	0.918
I ask friends or relatives for advice.	2.41	1.24	0.007	0.518
I consult a pediatric book.	3.24	1.39	0.003	0.245
I drive the child to the nearest hospital immediately.	3.30	1.25	0.201	0.905
I call an ambulance.	3.84	1.28	0.364	0.816
**Scenario 2:**				
**Imagine that at 8:00 p.m. your child suddenly shows an itchy rash on the body and complains about severe headache. The private practices of your general practitioner and your pediatrician are already closed. You do not know why your child feels that way. You want to visit your family doctor in the morning, if your child does not feel better by then, but you also want to search online about your child‘s condition.** Please indicate the likelihood for using each of the following online platforms (each: low = 1 to high = 4).				
Netdoktor	2.80	0.85	0.276	0.183
Internet forum	2.11	0.85	0.015	0.663
Homepage of a federal ministry	1.99	0.97	0.252	0.304
Gesundpedia	1.90	0.84	0.155	0.882
DocCheck Flexikon	1.73	0.82	0.885	0.480
Social media channel	1.54	0.77	0.236	0.539

Notes: # Mann–Whitney U tests for the dichotomous variables age groups (young vs. older) and gender (males vs. females).

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
