# Peer review of "Mobile and Online Health Information: Exploring Digital Media Use among Austrian Parents"

_ijerph, 2020, doi:10.3390/ijerph17176053_

Round 1

Reviewer 1 Report

The authors generally addressed my questions and concerns well. I only have two suggestions left. First, I still don’t know why the authors decided to dichotomize the age variable, instead of treating it as a continuous variable. Please clarify or consider conduct an additional analysis of treating age as a continuous variable to avoid the concern of model extrapolation. Second, the p-value cut-off point should be .05, not .5.

Author Response

Response to Reviewer 1 Comments

Point 1:

The authors generally addressed my questions and concerns well. I only have two suggestions left. First, I still don’t know why the authors decided to dichotomize the age variable, instead of treating it as a continuous variable. Please clarify or consider conduct an additional analysis of treating age as a continuous variable to avoid the concern of model extrapolation. Second, the p-value cut-off point should be .05, not .5.

Response 1:

We thank Reviewer 1 for the favorable evaluation of our manuscript.

As for the age, we used median splits to generate a dichotomous variable in order to be able to interpret results for a younger and an older segment of the study sample. This is especially useful when analyzing more than one subgroup such as the dichotomous variables age groups (young vs. older) and gender (males vs. females).

This is convenient and commonly done, exemplarily when investigating aspects of digital age, which was beyond the scope of this study due to restrictions in the sample characteristics.

As for the p values, we are sorry for this and corrected the cut-off point accordingly.

Reviewer 2 Report

Thanks for the revisions of the earlier manuscript. Just a few corrections remain.

These can be easily done in a few minutes.

Page 2, Line 56, and other repetitions throughout the manuscript:

The term “feasibility study” should be changed to “exploratory study.”

Page 2, Line 73: Change “barrier-freely” to “barrier-free”

Page 3, Line 92:

Besides of the socio-demographic characteristics as mentioned above . . .

Besides the socio-demographic characteristics mentioned above . . .

Page 3, Line 99:

A multiple choice question asked participants to indicate whether having already searched 99 online for information on family policy benefits (yes vs. no).

A multiple-choice question asked participants to indicate whether they had already searched online for information on family policy benefits (yes vs. no).

Page 4, Line 139:

We conducted all reported statistical analyses using SPSS Version 26.0 (SPSS Inc., Chicago, IL, USA)

This error was corrected in my previous review. SPSS is now published by IBM. See

https://www.ibm.com/analytics/spss-statistics-software

Author Response

Response to Reviewer 2 Comments

Point 1:

Thanks for the revisions of the earlier manuscript. Just a few corrections remain.

These can be easily done in a few minutes.

Response 1:

We thank Reviewer 1 for the favorable evaluation of our manuscript.

Point 2:

Page 2, Line 56, and other repetitions throughout the manuscript:

The term “feasibility study” should be changed to “exploratory study.”

Page 2, Line 73: Change “barrier-freely” to “barrier-free”

Page 3, Line 92:

Besides of the socio-demographic characteristics as mentioned above . . .

Besides the socio-demographic characteristics mentioned above . . .

Page 3, Line 99:

A multiple choice question asked participants to indicate whether having already searched 99 online for information on family policy benefits (yes vs. no).

A multiple-choice question asked participants to indicate whether they had already searched online for information on family policy benefits (yes vs. no).

Page 4, Line 139:

We conducted all reported statistical analyses using SPSS Version 26.0 (SPSS Inc., Chicago, IL, USA)

This error was corrected in my previous review. SPSS is now published by IBM. See

https://www.ibm.com/analytics/spss-statistics-software

Response 2:

We are thankful for indicating the typos and spelling mistakes, which we have all corrected in our revised version of the manuscript. Specifically, we cite SPSS as follows:

“We conducted all reported statistical analyses using SPSS Statistics for Windows, Version 26.0 (Armonk, NY, USA, IBM Corp.).”

Reviewer 3 Report

Thank you for the opportunity to review this paper. This was a timely read, especially given the current climate and global push towards more virtual healthcare service provisions. You’ve made a case for the study from your introduction that clearly describes why the study is needed within the specified context. 

There were no issues of significant concern with respect to the research methodology, and I was generally satisfied that the chosen approach was appropriate to answer the research questions.

Since you report a snowball sampling approach, it will be useful to provide details of the initial online forums, blog etc that the survey link was posted. This will be informative for other researchers to judge appropriateness and also replicate study in other settings if needed.

Also, there are a few typos on line 146 – 147 concerning the n numbers quoted, please review and correct as appropriate. Other than these, I think you paper is very well presented.

Best wishes

Author Response

Response to Reviewer 3 Comments

Point 1:

Thank you for the opportunity to review this paper. This was a timely read, especially given the current climate and global push towards more virtual healthcare service provisions. You’ve made a case for the study from your introduction that clearly describes why the study is needed within the specified context. 

There were no issues of significant concern with respect to the research methodology, and I was generally satisfied that the chosen approach was appropriate to answer the research questions.

Since you report a snowball sampling approach, it will be useful to provide details of the initial online forums, blog etc that the survey link was posted. This will be informative for other researchers to judge appropriateness and also replicate study in other settings if needed.

Also, there are a few typos on line 146 – 147 concerning the n numbers quoted, please review and correct as appropriate. Other than these, I think you paper is very well presented.

Best wishes

Response 1:

We thank Reviewer 1 for the favorable evaluation of our manuscript.

Attempting a true snowball sampling approach, we actually started with our personal social media sites for recruitment. In response to this request, we modified the following phrase:

Attempting a snowball sampling approach, we sent the non-personalized link to the online questionnaire to potential survey participants via email invitations and posted it to social media sites and groups (Facebook, WhatsApp) as well as specific parenting blogs, with the request to circulate the link in the respondents` own network of friends and family.”

We also added the following clause to the limitations section:

“…limiting generalizability of study results and representativeness of the study sample”.

Many thanks for pointing out the typos, which presumable root in using the track change function in a previous version of the manuscript.

Reviewer 4 Report

Overall, the manuscript seems to alternate between a focus on childcare to a more general overview about the use and predisposition to use health apps, in general. The only aspect of the study that makes a connection with childcare is the consideration of the scenarios and a restriction of the considered participants to those having children. However, aspects such as children's ages as a main factor for the analysis or the consideration of health apps supporting childcare, which would probably make sense given the scope of the questionnaires, is not presented.

The final conclusion drawn from the study is, in my humble opinion, not substantiated by any part of the results or study. To conclude something about affectiveness and efficiency in personal contacts in the doctor-patient-child triangle appears without any clear rationale to support it (last sentence in the abstract). This, somehow, seems to hint that people should resort more to the doctors, but if the authors wanted to take the discussion into doctoor patient relationship shouldn't they be looking into those services and apps that make an attempt to perform it? So, it seems that this study was not designed for such a purpose.

With the data they have, as far as I could grasp, they cannot assert that the participants prefer the web rather than the pediatrician, for example. So taling about doctor-patient relationships and, then, stating something like "The participants indicated to prefer online information seeking to any other option when their child was sick" needs to be properly clarified, regarding the extent of its applicability, and is rather limited by the fact that the "any other option" was a short list not including personal contact with a doctor. The question that remains unanswered is "will they prefer to have a look online and do they trust it? Or is it the search online just a resort that is favoured given the conditions of the scenario? Do I look online and make a decision or do I look online to decide my next exploration step?

In scenario 2, a fixed list of sites is provided. Why is not there an option for reporting on other sources? It is difficult to understand that the probably common path through Google (or any other general search engine) is not contemplated. In fact, the results, it seems, do not strongly favor any of the presented choices. So, why force a choice and not leave room to understand what is actually used?

The different limitations imposed by the scenarios, e.g., the time of 8 p.m.: wouldn't this preclude contacting friends or relatives, at least in a first moment, so not to disturb them at this late hour?

While a focus on health apps use is covered by parts of the questionnaire, a connection between this trend and the use of health apps specific for childcare contexts is not tackled or even mentioned. Do parents use specific apps for this purpose? This is not covered by the presented data and the limited scope of the choices provided for scenario 2 does not help.

Any reason why the group of parents with younger children (or just one) is not specifically explored? What happens to parents with children below a certain age? Why is data not explored to consubstantiate that "There is evidence that especially first-time parents might differ regarding their need for professional advice" (line 277) Not all first-time parents need to be the younger participants. The more common use of technology by younger participants may work as a confound, here.

Finally, one of the aspects I miss is a clear rationale of which outcomes of this study drive which plans or paths for future research providing a clear rationale to support that "the findings of this study prepare the ground for future research, which should be refined in more complex research approaches in a large, representative study population to elucidate further mechanisms of information retrieval among German-speaking parents and guardians (lines 302-205). Whats does this actually entail?

Other comments:

- Please check (I may have missed it), but as far as I could grasp, no reference to Table 1 is provided, in the text.
- The list of sites given as options appears strangely typeset. Some connection word or sentence missing?
- Line 147, typos for the values of n

Author Response

Response to Reviewer 4 Comments

Point 1:

Overall, the manuscript seems to alternate between a focus on childcare to a more general overview about the use and predisposition to use health apps, in general. The only aspect of the study that makes a connection with childcare is the consideration of the scenarios and a restriction of the considered participants to those having children. However, aspects such as children's ages as a main factor for the analysis or the consideration of health apps supporting childcare, which would probably make sense given the scope of the questionnaires, is not presented.

Response 1:

We thank the Reviewer for providing feedback on our manuscript and are grateful for the insightful comments on and valuable improvements to our paper.

We agree that due to ethical and practical reasons, we aimed at assessing and reported the parental perspective only. We used information on health app use and readiness for using health apps as a base to underline the findings of the scenarios. In response to this request, we refined the title accordingly:

“Mobile and Online Health Information: Exploring Digital Media Use among Austrian Parents”

Please be aware that average age of the respondents` children has already been reported. We found that - as expected - older age of parents correlated with age of the children. Parents with more than one child might also have more confidence when a further child gets sick. On the other hand, parents with a chronically ill child might act differently. This exploratory study aimed at surveying the parents, but we agree that it would make sense to integrated child age when conducting a further larger study on a representative study sample. Alternatively, age of parents could be restricted in data collected or analysis, e.g. only females in reproductive age.

In response to this request, among others, we added the following paragraph to the limitations section:

“As expected, age of parents correlated with age of the children. Parents with more than one child might also have more confidence when a younger sibling gets sick. On the other hand, parents with a chronically ill child might act differently. Thus, aspects of child age and other child-specific characteristics might be integrated in further studies.”

Point 2:

The final conclusion drawn from the study is, in my humble opinion, not substantiated by any part of the results or study. To conclude something about affectiveness and efficiency in personal contacts in the doctor-patient-child triangle appears without any clear rationale to support it (last sentence in the abstract). This, somehow, seems to hint that people should resort more to the doctors, but if the authors wanted to take the discussion into doctoor patient relationship shouldn't they be looking into those services and apps that make an attempt to perform it? So, it seems that this study was not designed for such a purpose.

Response 2:

This statement refers to our findings and the related discussion on aspects of e.g. parental education, doctor-patient-relationship, counseling, subgroup differences etc. However, we agree and defuse this statement to hopefully increase clarity.

“Pediatricians are best suited to decide what treatment fits to the child or its current medical condition, but they nowadays face increasing numbers of pre-informed parents seeking health information online. Provision of targeted parental education and guidance through the online information jungle could effectively empower parents and smooth personal and digital contacts in the delicate doctor-parent-child triangle.”

Point 3:

With the data they have, as far as I could grasp, they cannot assert that the participants prefer the web rather than the pediatrician, for example. So taling about doctor-patient relationships and, then, stating something like "The participants indicated to prefer online information seeking to any other option when their child was sick" needs to be properly clarified, regarding the extent of its applicability, and is rather limited by the fact that the "any other option" was a short list not including personal contact with a doctor. The question that remains unanswered is "will they prefer to have a look online and do they trust it? Or is it the search online just a resort that is favoured given the conditions of the scenario? Do I look online and make a decision or do I look online to decide my next exploration step?

In scenario 2, a fixed list of sites is provided. Why is not there an option for reporting on other sources? It is difficult to understand that the probably common path through Google (or any other general search engine) is not contemplated. In fact, the results, it seems, do not strongly favor any of the presented choices. So, why force a choice and not leave room to understand what is actually used?

The different limitations imposed by the scenarios, e.g., the time of 8 p.m.: wouldn't this preclude contacting friends or relatives, at least in a first moment, so not to disturb them at this late hour?

While a focus on health apps use is covered by parts of the questionnaire, a connection between this trend and the use of health apps specific for childcare contexts is not tackled or even mentioned. Do parents use specific apps for this purpose? This is not covered by the presented data and the limited scope of the choices provided for scenario 2 does not help.

Response 3:

Please note that we deliberately used the scenario technique to circumvent single items or scales with lot of room for interpretation. So, the statement mentioned refers to the finding of the scenario, where explicitly no personal contact to general practitioner or pediatrician was an option, whereas we found in the pre-test, that in case of an emergence, people (at least in Austria) would call their closest friends or relatives at any time (even at midnight), but we decided to choose the time of 8 p.m. for convenience reasons, as at this time, doctor`s offices are already closed, but most people are still awake. To increase clarity, we no provide the following in the abstract:

“The participants indicated to prefer online information seeking to any other option in a scenario describing that their child would be sick at after-work hours, with social media channels being the least preferred source of online information.”

As for the comment on “Dr. Google”, we agree and thus also mentioned this common search strategy in our manuscript. In accordance with the literature, we assumed that almost all participants would then indicate using Google or other general search engines. However, in response to this comment, we modified the respective phrases to increase transparency of the underlying theoretical concepts and also suggested to use mixed-method approaches in future studies to also cover qualitative evaluations.

Point 4:

Any reason why the group of parents with younger children (or just one) is not specifically explored? What happens to parents with children below a certain age? Why is data not explored to consubstantiate that "There is evidence that especially first-time parents might differ regarding their need for professional advice" (line 277) Not all first-time parents need to be the younger participants. The more common use of technology by younger participants may work as a confound, here.

Response 4:

Some of the aspects on age raised here have been already covered in Point 3. Please note that the additional aspect of digital age has already been discussed. In response to this issue, we now state that socio-cultural characteristics also include digital age. Detailed subgroups analysis on various age groups is beyond the scope of this study due to limited subgroups size and should be integrated in further research, as suggested.

Point 5:

Finally, one of the aspects I miss is a clear rationale of which outcomes of this study drive which plans or paths for future research providing a clear rationale to support that "the findings of this study prepare the ground for future research, which should be refined in more complex research approaches in a large, representative study population to elucidate further mechanisms of information retrieval among German-speaking parents and guardians (lines 302-205). Whats does this actually entail?

Response 5:

Given the modifications done in the limitations section as described above (e.g. aspects of age), we hope that the last concluding sentence makes more sense.

Point 6:

Other comments:

- Please check (I may have missed it), but as far as I could grasp, no reference to Table 1 is provided, in the text.
- The list of sites given as options appears strangely typeset. Some connection word or sentence missing?
- Line 147, typos for the values of n

Response 6:

- Reference to Table 1 was provided in the paragraph preceding the table.

- We checked for those typos.

- Many thanks for pointing out the typos, which presumable root in using the track change function in a previous version of the manuscript.

Round 2

Reviewer 4 Report

I thank the reviewers for their effort in providing a revised version of their manuscript addressing some of my concerns and questions. My main concern was that there was an unrealistic overtone of what could be concluded from the protocol/data.

Overall, I consider that, by toning down some of the conclusions, and rephrasing some parts, the manuscript got more realistic and clear about the provided contributions.

Some minor suggestions that, at the authors' discretion, may improve clarity:

Eventually, I would explicitly state "we found in the pre-test, that in case of an emergence, people (at least in Austria) would call their closest friends or relatives at any time (even at midnight),", in the manuscript (sorry if I failed to spot it), since this would strengthen the value of the conclusions drawn from the scenario-based results.

In my humble opinion, the authors' do a better job in conveying the rationale on how to connect their results with the doctor-patient relationship by introduciong the access to online information as a point that often mediates this relationship. This is not particularly true for the end of the conclusions and I would suggest changing the conclusions' text to be more aligned with this rationale.

Author Response

Response to Reviewer 4 Comments

Point 1:

I thank the reviewers for their effort in providing a revised version of their manuscript addressing some of my concerns and questions. My main concern was that there was an unrealistic overtone of what could be concluded from the protocol/data.

Overall, I consider that, by toning down some of the conclusions, and rephrasing some parts, the manuscript got more realistic and clear about the provided contributions.

Some minor suggestions that, at the authors' discretion, may improve clarity:

Eventually, I would explicitly state "we found in the pre-test, that in case of an emergence, people (at least in Austria) would call their closest friends or relatives at any time (even at midnight),", in the manuscript (sorry if I failed to spot it), since this would strengthen the value of the conclusions drawn from the scenario-based results.

Response 1:

We thank Reviewer 4 for favorably evaluating the revised version of our manuscript.

In response to this request, we include the following statement in the text, as originally it was solely part of the response to the Reviewer´s comment. However, we agree that this information would be useful for the readership.

“In the pre-test, we explicitly verified that in case of emergency, people in Austria would commonly not hesitate to call their closest friends or relatives at any time (even at midnight). So, we chose the time of 8 p.m. for convenience reasons, as at this time, doctors` offices are already closed, but most people are still awake.”

Point 2:

In my humble opinion, the authors' do a better job in conveying the rationale on how to connect their results with the doctor-patient relationship by introduciong the access to online information as a point that often mediates this relationship. This is not particularly true for the end of the conclusions and I would suggest changing the conclusions' text to be more aligned with this rationale.

Response 2:

We agree and modified the conclusion section accordingly. We now state the following:

“In today’s digitalized world, parents more and more rely on web content when searching for child health information. With the smartphone in everybody’s pocket, this trend is very likely to be continued in the future. Pediatricians are trusted experts in giving evidence-based advice on childcare and treatment options, but an increasing numbers of pre-informed parents at point of care affect the traditional doctor-patient relationship in the digital age. Thus, mutual understanding and in-depth knowledge of information retrieval habits in general and target-group specifically are essential in the per se delicate and complex interactions expectable in the doctor-parent-child triangle. For that, legal support and pro-family policies should integrate this newly established web-based parental empowerment and health literacy in family and health policy and decision-making processes.”

This manuscript is a resubmission of an earlier submission. The following is a list of the peer review reports and author responses from that submission.

Round 1

Reviewer 1 Report

The study examined the health app use among Austrian parents. Although this is a needed topic, I have several major and minor concerns with publishing this manuscript.

First and foremost, I really don’t know what meaningful conclusion could be drawn with a convenience sample of only 90 participants. Such small sample size not only renders the inferential statistical results unreliable, but also not representative at all in describing the health app use of Austrian parents, especially as a nonprobabilistic sample.

The manuscript is poorly organized, starting from a weak rationale. For instance, what’s the rationale of measuring “patient/public readiness assessment for telehealth tool (PRAT)”?  This variable has not been mentioned in the literature review at all, but all of a sudden appeared in the methods section, similar with benefits and risk.

The literature review should have been more relevant. For instance, what’s the purpose of describing the qualitative study by Van Der Gugten et al., which focused on Dutch parents? Are the Dutch population comparable to Austrian population?

There’re numerous problems in the methods section.

  • First, the authors don’t need to present the details of data collection (e.g., “Collected data was only accessible for research team members”, “We did not offer incentives for study participation.”) that were presented in the IRB protocol in the manuscript.
  • Second, the description of the methods is by no means clear. For instance, in which scale do these two questions come from—“benefits or anticipated benefits/risks” and “practitioner mediated liaison for telehealth programs”? How were the two scenarios determined? Also, since they overlapped significantly, when I read the manuscript, I was wondering why the authors included two scenarios. It would make the argument clearer to move the description of the two scenarios on p.10 to the methods section.
  • Third, the descriptive and inferential statistical analyses were conducted incorrectly. Why did the author dichotomize age? What is relative frequency, and how was it analyzed? What’s the meaning of conducting chi-square of age and having children? Descriptive statistics were not conducted meaningfully, such as describing the average number of children. Presenting the frequency of participants having 1 to 4 children would be sufficient.
  • The inferential statistics are also very confusing. I don’t know what analysis the authors did by reading “In our study sample, we did not find statistically significant gender differences according to age, place of living, amount of children as well as health app use, which was high with 7 %”. Were age, place of living, amount of children as well as health app use dependent variables? If so, what’s the meaning of predicting age, place of living, and amount of children by gender?
  • Finally, 0.6 is not a good threshold for Cronbach's alpha. An acceptable reliability requires a Cronbach’s alpha .7 or higher. That said, the reliability of the readiness is low

Finally, the theoretical and practical contributions of the current study are unclear in the discussion.

Several grammar mistakes and typos are present as well. Such as “A higher education level of parents as well as a younger age and an acute disease of the child predisposed for online health information seeking”, and “convenient sample”.

Reviewer 2 Report

Detailed comments are in the attached file.
